# Clovamide and Its Derivatives—Bioactive Components of *Theobroma cacao* and Other Plants in the Context of Human Health

**DOI:** 10.3390/foods13071118

**Published:** 2024-04-06

**Authors:** Joanna Kolodziejczyk-Czepas

**Affiliations:** Department of General Biochemistry, Faculty of Biology and Environmental Protection, University of Lodz, Pomorska 141/143, 90-236 Lodz, Poland; joanna.kolodziejczyk@biol.uni.lodz.pl

**Keywords:** clovamide, cocoa beans, phenolamide, antioxidant, anti-inflammatory, neuroprotective

## Abstract

Clovamide (*N*-caffeoyl-L-3,4-dihydroxyphenylalanine, *N*-caffeoyldopamine, *N*-caffeoyl-L-DOPA) is a derivative of caffeic acid, belonging to phenolamides (hydroxycinnamic acid amides). Despite a growing interest in the biological activity of natural polyphenolic substances, studies on the properties of clovamide and related compounds, their significance as bioactive components of the diet, as well as their effects on human health are a relatively new research trend. On the other hand, in vitro and in vivo evidence indicates the considerable potential of these substances in the context of maintaining human health or using them as pharmacophores. The name “clovamide” directly derives from red clover (*Trifolium pratense* L.), being the first identified source of this compound. In the human diet, clovamides are mainly present in chocolate and other cocoa-containing products. Furthermore, their occurrence in some medicinal plants has also been confirmed. The literature reports deal with the antioxidant, anti-inflammatory, neuroprotective, antiplatelet/antithrombotic and anticancer properties of clovamide-type compounds. This narrative review summarizes the available data on the biological activity of clovamides and their potential health-supporting properties, including prospects for the use of these compounds for therapeutic purposes.

## 1. Introduction

Among a variety of components of the human diet, *Theobroma cacao*-derived products (such as cocoa beans, cocoa powder and cocoa-based beverages) have maintained a special position since ancient times. Their long history includes a variety of applications, e.g., as a drink to maintain health and to strengthen warriors, as a common currency throughout Mesoamerica, and as a remedy [1,2]. Based on the results of contemporary studies, the health benefits of cocoa have been attributed to a wide spectrum of bioactive components, including (poly)phenolic compounds such as catechins, anthocyanins, proanthocyanidins, flavonols, lignans, and stilbenes, as well as phenolic acids and their derivatives [3,4].

Clovamide, i.e., (2S)-3-(3,4-dihydroxyphenyl)-2-[[(E)-3-(3,4-dihydroxyphenyl)prop-2-enoyl]amino]propanoic acid), also known as *N*-caffeoyl-L-3,4-dihydroxyphenylalanine, *N*-caffeoyldopamine or *N*-caffeoyl-L-DOPA, is a derivative of caffeic acid (3,4-dihydroxycinnamic acid) and an amide isostere of rosmarinic acid. It belongs to a large family of plant phenolamides, comprising compounds derived from the association of phenolic acids with aliphatic or aromatic amines. Due to their many biological activities (Figure 1), phenolamides have garnered scientific interest in the context of their administration for health-improving purposes and their use in the cosmetics industry [5]. Chemically, they are classified as hydroxycinnamic acid amides (HCAAs), a diverse group of specialized plant phenylpropanoid metabolites that are involved in maintaining plant tolerance to abiotic and biotic stress [6]. The metabolic precursors for the natural synthesis of HCAAs in plants are tyrosine and its derivatives, such as tyramine and dopamine. The key stage in their biosynthesis is the formation of an amide bond between the carboxylic groups of phenylpropenoic acids and the amino groups of amino acids [7]. For the chemical synthesis of clovamide, *trans*-caffeic acid and L-DOPA methyl ester are used [8]; however, the synthesis of clovamide and its analogues using engineered microbial hosts, i.e., *Saccharomyces cerevisiae* and *Lactococcus lactis*, has been described as well [9].

The clovamide name derives from the red clover (*Trifolium pratense* L.), i.e., the first identified plant source of this compound [10], and the collective term “clovamides” includes both *cis*- and *trans*-clovamide as well as other clovamide-type substances such as deoxyclovamide, dideoxyclovamide, clovamide methyl ester and other derivatives (Figure 2).

Although this compound was identified as early as the 1970s, scientific interest in its biological activity was marginal at that time. However, recent decades have provided numerous reports dealing with newly identified plant sources as well as the beneficial properties of clovamide and its related compounds (Figure 3). This work is focused on clovamides as a group of bioactive components of *Theobroma cacao* L. (cocoa) beans and other plants. It presents the available data on the biological activity of clovamide and its derivatives in the context of their effects on human physiology (i.e., health-supporting actions) and prospects for use in therapeutics.

## 2. Plant Sources of Clovamides

Although HCAAs are widely distributed in the plant kingdom, so far, the presence of clovamides has been revealed only in members of several genera, e.g., *Theobroma cacao* L. [11], *Theobroma grandiflorum* (Willd. ex Spreng.) K. Schum. [12], some *Trifolium* (clover) species [13,14], *Vernonia fastigiata* Oliv. & Hiern [15], *Dalbergia melanoxylon* Guill. & Perr. [16], *Acmella oleracea* (L.) R.K. Jansen [17], *Acmella ciliata* (Kunth) Cass. [18], *Dichrostachys cinerea* (L.) Wight et Arn [19], *Ceiba pentandra* (L.) Gaertn. [20], *Zinnia elegans* Jacq. [21] and *Urtica dioica* L. [22].

In the human diet, the main source of clovamides is chocolate and other cocoa-based products [23,24,25] (Table 1), but these compounds have been also identified in *T. grandiflorum* (cupuassu) fruits (clovamide content established to be 23.03 mg/g dry weight of the phenolic extract) [12] and Robusta coffee (*Coffea canephora* Pierre ex A. Froehner, syn. *Coffea robusta* L. Linden) beans [26].

Furthermore, a high content of clovamides has been found in the aerial parts of some species of clovers. Whatever their functions as fodder plants, some clover species are also used as medicinal and edible plants. The most known is the red clover (*T. pratense*), a component of herbal preparations and dietary supplements administered to perimenopausal women [27,28] and an ingredient in vegan cuisine. A comparative phytochemical profiling of 57 clover specimens revealed that some of them were particularly rich in clovamides (data summarized in Table 2 and Figure 4). The highest clovamide content was detected in *Trifolium pallidum* and amounted to 12.94 mg/g of d.w., corresponding to 36% of the total content of polyphenolic compounds detected in the examined herbal material originating from this plant [29]. Later studies on clovers confirmed that the *Trifolium* genus contains species that can be a rich source of clovamide and its methyl ester, the main clovamide derivative detected in plant tissues. A clovamide content exceeding 21% of the plant extract dry mass was found in extracts from *Trifolium clypeatum*, *T. obscurum* and *T. squarrosum,* extracted with 80% methanol [30]. Furthermore, different clovamide-type compounds such as *cis*-clovamide and *trans*-clovamide, together with caffeoyl-*N*-tyrosine, coumaroyl-*N*-tyrosine and feruloyl-*N*-dopamine, were identified and quantified in the aforementioned study. It should be noted, however, that the clovamide content may vary significantly in plant material originating from different species or even subspecies. In the aerial parts of *T. pratense*, *trans*-clovamide is synthesized and accumulated at a relatively high level, being attained at up to around 1% of dry matter, and its content is dependent on the genotype [31]. In an 80% methanol extract from *T. pratense*, the clovamide content attained was 15.6 mg/g of d.w., whereas in the same type of extract isolated from *T. pratense* subsp. *nivale*, the clovamide content was 8.2 mg/g of d.w. [32].

## 3. Bioavailability and Metabolism of Clovamide-Type Compounds

Clovamides are characterized by a low threshold for perceptible astringent effects (concentrations > 10 µmol/L), and therefore their ingestion may evoke astringent effects in the mouth [33]. Contrary to the well-established blood plasma levels of flavonoids and phenolics acids or their metabolites [34,35,36], data on the bioavailability and metabolism of clovamide(s) are limited. Given the ample evidence on other low-molecular-weight polyphenols (including phenolic acids), it can be assumed that the maximal plasma levels of clovamides/their metabolites range from nanomoles to a few micromoles per liter. The intestinal permeability of rosmarinic acid, which has a significant structural similarity to clovamide, was established to attain <1% of its intake volume [37]. After the peroral intake of 200 mg of extract from *Perilla frutescens* L., the plasma concentration of rosmarinic acid (both in conjugated and non-conjugated forms) was 1.15 ± 0.28 μmol/L [38]. The maximal plasma concentrations of related compounds such as chlorogenic, caffeic and ferulic acid were established as 0.26, 0.96 and 0.03 μmol/L, respectively, and their urinary excretion amounted to 0.3, 10.7, and 27.6% of their intake, respectively [34]. According to one of very few reports on the clovamide(s) metabolism, at 30 min after the oral administration of *N*-caffeoyltyramine to animals (at a dose of 0.1 mg/30 g of body weight), its plasma concentration amounted to 50 nM [7]. In mice treated with a single dose of a clovamide-containing extract from cupuassu fruit (5 mg of catechin equivalents/kg of b.w.), the clovamide content decreased and reached the next sections of the digestive tract, attaining the maxima at 0.5 h in the stomach, 1 h in the small intestine, 2 h in the caecum and 3 h in the colon [12].

It has been suggested that *N*-phenylpropenoyl-L-amino acids undergo microbial degradation, leading to their transformation into phenolic acids [39,40]. Other studies on the absorption of phenylpropanoid conjugates with amino acids have indicated that contrary to other groups of plant (poly)phenols, after the consumption of a cocoa-based drink, these compounds are metabolized neither via their *O*-glucuronides nor their *O*-sulfates. These findings indicate that the *O*-methylation or reduction in the phenylpropanoid part may play an important role in the metabolic transformations of these compounds [33].

## 4. Biological Activity of Clovamides

### 4.1. Antioxidant Action

Following the discovery that oxidative stress is one of key factors involved in the development and progression of many societal diseases [41,42,43,44], antioxidant action has become one of the most frequently studied properties of natural and synthetic substances [45,46,47]. The mechanisms of the antioxidant action of polyphenols (including phenolic acids) involve either hydrogen atom transfer (HAT), single-electron transfer (SET), sequential proton loss electron transfer (SPLET) or the chelation of transition metal ions [48]. The clovamide-type compounds seem to share all of the aforementioned properties. However, recent liquid chromatography–mass spectrometry (LC-MS) analyses have indicated that HAT is the most preferred mechanism of clovamide antioxidant action. Clovamide is a better scavenger of hydroxyl radicals than peroxyl radicals, and its mode of radical scavenging is higher in non-polar medium than in the polar milieu [49]. The high free-radical-scavenging potency of clovamide is attributed to the presence of hydroxyl groups, especially those in catechol moieties. Moreover, two catechol moieties are considered to have a more important role in the scavenging activity than the other hydroxyl groups [23].

The history of studies on clovamide and its derivatives as antioxidants began in the 1990s, when the anti-lipoperoxidative activity of *T. cacao* polyphenols, including clovamide and deoxyclovamide, was revealed. Clovamide has been found to be a more efficient antioxidant than other polyphenols in linoleic acid oxidation assays, surpassing even the activity of epicatechin and quercetin (the order of antioxidant efficiency is as follows: clovamide > epicatechin > catechin > quercetin > quercetin 3-glucoside, quercetin 3-arabinoside, and dideoxyclovamide (*N*-*trans*-*p*-coumaroyl-L-tyrosine)) [11]. The inhibitory effects of the clovamide and phenolics obtained from cocoa beans on lipid peroxidation in liposomal systems were also confirmed by another research group [50].

Despite a significant increase in the number of studies on the antioxidant activity of clovamides in the last two decades, the contribution of this type of clovamide activity to overall human health still remains difficult to estimate. Current knowledge of the anti-radical and antioxidant potential of clovamides is based only on in vitro studies, including synthetic and physiological radical scavenging tests and lipid peroxidation assays (Table 3). Moreover, numerous papers have questioned the in vivo significance and health benefits of many phytochemicals with antioxidant effects evidenced in vitro [51,52,53,54,55]. An important issue is also the thermal processing of the food, bearing with it the risk of a partial loss of bioactive substances. For instance, measurements of the superoxide radical (O_2_^•−^)-scavenging ability of clovamide and clovamide-containing cocoa bean samples conducted in the Rotating Ring-Disk Electrode (RRDE) electrochemical system revealed a positive correlation between the clovamide content and antioxidant activities of beans. However, the roasting of cocoa beans reduced both the clovamide content (by 14.3%) and the overall antioxidant activity (by 18.2%) [23].

On the other hand, reports in the literature indicate the considerable antioxidant potential of clovamide(s) when compared to many other well-described phenolic antioxidants of plant origin (Table 3). Clovamide displayed an ability to scavenge 2,2-diphenyl-1-picrylhydrazyl radicals (DPPH^•^) that is comparable to that of rosmarinic and caffeic acids, and even exceeded the effectiveness of butylated hydroxyanisole, a synthetic antioxidant that is used as a food additive (E320) [56]. In the prevention of the oxidation of sunflower oil triacylglycerols, caffeoyldopamine (clovamide), cinnamoyldopamine, *p*-coumaroyldopamine, feruloyldopamine, sinapoyldopamine, caffeoyltyramine and caffeoyltryptamine displayed a comparable or higher antioxidant activity than caffeic acid (a reference compound). Moreover, in that test, clovamide was the most efficient one [57]. A few years ago, a series of clovamide derivatives were synthesized using a coupling reaction between L-phenylalanine (L-DOPA; L-dopamine) and cinnamic acids derivatives. The DPPH^•^-scavenging efficiencies of some of the obtained clovamide esters were found to have an EC_50_ ranging from 1.55 to 4.23 μg/mL, similar to the efficiency observed for quercetin (IC_50_ = 1.20 μg/mL) [58]. In other work, two ester derivatives of clovamide, i.e., *N*-caffeoyl-L-tyrosine and *N*-caffeoyl-L-dihydroxyphenylalanine, were examined in the context of their usefulness as potential food or cosmetic antioxidant additives. Both esters protected the soybean oil components against autoxidation, with activity at a level comparable to or higher than the activity of known antioxidants such as α-tocopherol and ascorbic acid [59].

Since oxidative stress is involved in many disorders of the cardiovascular system [60,61,62], the antioxidant action of many natural compounds has been examined using blood cells, plasma components or other experimental models related to the cardiovascular physiology. This search for a link between the biological activity of plant components of the human diet and the functioning of the circulatory system also includes studies on clovamides. In monocytes and H9c2 cardiomyocytes exposed to oxidative stress, clovamide and cocoa bean extract (though to a lesser extent) have displayed significant anti-inflammatory and antioxidant activity. The examined substances decreased the generation of O_2_^•−^, which is one of the most important reactive oxygen species (ROS) formed in the human body. A preliminary evaluation of the effects on cardiomyocytes revealed a reduction in ROS generation and apoptosis by clovamide and epicatechin [63]. At a micromolar concentration (3 μM), clovamide was also found to protect the cardiac progenitor cells (CPCs) isolated from human heart biopsies against the oxidative damage induced by hydrogen peroxide. The treatment of CPCs with clovamide significantly decreased the H_2_O_2_-induced generation of ROS, lipid peroxidation and apoptosis [64]. The clovamide-rich fraction (containing 0.58 g/g d.w. of clovamide and 0.16 g/g d.w. of its methyl ester) obtained from *Trifolium pallidum* was also found to reduce peroxynitrite (ONOO^−^)-triggered damage to the lipid and protein components of human plasma and platelets in vitro [65]. A reduction in the ONOO^−^-mediated modification of blood plasma proteins and lipids by *trans*-clovamide and clovamide-rich extracts isolated from three *Trifolium* species (i.e., *T. clypeatum* L., *T. obscurum* Savi and *T. squarrosum* L.) has been also described. Interestingly, all of those substances (i.e., clovamide and extracts) displayed the ability to limit the oxidative rather than the nitrative damage to blood plasma components [30].
foods-13-01118-t003_Table 3Table 3Free-radical-scavenging and antioxidant activities of clovamide and related compounds. Clovamide activity was evaluated based on the half maximal effective concentration (EC_50_; in the DPPH^•^, ONOO^−^ and O_2_^•−^-scavenging tests) or the half maximal inhibitory concentration (IC_50_; in the β-carotene bleaching assay).Experimental ModelEC_50_ or IC_50_ Values Established for ClovamideEC_50_ or IC_50_ ValuesEstablished for Reference CompoundsReferencesDPPH^•^ scavengingClovamide: 2.65 μg/mLCaffeic acid: 2.93 μg/mLEpicatechin: 3.11 μg/mLGallic acid: 1.03 μg/mLRosmarinic acid: 2.49 μg/mLMyricetin: 1.95 μg/mLQuercetin: 1.99 μg/mLKaempferol: 4.26 μg/mLBHA: 8.18 μg/mLTrolox: 3.32 μg/mLOctyl gallate: 1.65 μg/mL[56]Clovamide: 4.9 μg/mLCaffeic acid: 4.2 μg/mLChlorogenic acid: 8.3 μg/mLTrolox: 5.7 μg/mL[30]Clovamide: 0.05 mol/molRosmarinic acid: 0.58 mol/molα-Tocopherol: 0.025 mol/molAscorbic acid: 0.025 mol/molL-dopamine: 0.095 mol/mol[59]ONOO^−^ scavengingClovamide: 19.3 μg/mLCaffeic acid: 15.0 μg/mLChlorogenic acid: 27.4 μg/mLTrolox: <5 μg/mL[30]Superoxide anion scavengingClovamide: 60 nmol/LRosmarinic acid: 95 nmol/Lα-Tocopherol: >10 000 nmol/LAscorbic acid: 700 nmol/LL-dopamine: 200 nmol/L[59]β-carotene bleachingClovamide: 0.02 mmol/Lα-Tocopherol: 0.08 mmol/LAscorbic acid: >0.09L-dopamine: >1.1 mmol/L[59]

### 4.2. Anti-Inflammatory Effects

Contrary to rosmarinic acid, a well-known natural anti-inflammatory agent [66,67,68], clovamides have been less described in the context of the alleviation of inflammatory processes. However, it is predicted that their presence may contribute to the health-promoting effects of some cocoa-containing food products. Comparative studies on the effects of clovamide and extracts derived from roasted and unroasted cocoa beans revealed the strongest anti-inflammatory effect in human monocytes incubated with clovamide. The molecular mechanisms of the anti-inflammatory action of clovamide involved inhibitory effects on the activation of the nuclear factor kappa-light-chain-enhancer of activated B cells (NF-κB), a reduction in pro-inflammatory cytokine release and a decrease in O_2_^•−^ generation [69]. Recent results from studies on zebrafish have demonstrated that N-*trans*-*p*-coumaroyltyrosine (NPCT) manages to inhibit the lipopolysaccharide (LPS)-induced inflammatory response, including the generation of nitric oxide and other ROS, as well as the migration of macrophages and neutrophils to the site of inflammation. The anti-inflammatory effects of NPCT were mediated via the signaling pathways dependent on the activation of NOD-like receptors (NLRs, the nucleotide-binding oligomerization domain-like receptors) and Toll-like receptors. Both of the aforementioned types of receptors are involved in cell reactions to pathogens and other molecules that are stressors, and thus the activators of the cell inflammatory response. At a biochemical level, the anti-inflammatory action of NPCT included the modulation of the gene expression of different elements of the inflammatory response, i.e., Toll-like receptor 4 (TLR4), the MyD88 protein, IRAK-4 (interleukin-1 receptor-associated kinase 4), NF-κB and its inhibitor kinase (IκB), the NLRP3 (nucleotide-binding domain, leucine-rich-containing family, pyrin domain-containing-3) inflammasome, caspase-1, the apoptosis-associated speck-like protein containing a caspase recruitment domain (ASC), as well as IL-1β and IL-6 interleukins [70].


### 4.3. Neuroprotective Effects

Despite many attempts to develop effective therapies for neurodegenerative diseases (NDs), this group of disorders remains a leading cause of death worldwide [71], and compounds of plant origin are considered one of the most important research trends in studies on neuroprotective substances. According to ethnomedicinal data, over 1300 plants have been established to display at least one bioactivity that may have therapeutic relevance for the treatment of neurodegenerative diseases [72]. Moreover, the positive effect of cocoa polyphenols on memory and executive functions has been reported [73,74]. The current state of the art does not allow exactly how the clovamide content contributes to these beneficial effects to be verified; however, several studies have evidently revealed the neuroprotective potential of clovamides. Results from comparative studies on the neuroprotective properties of clovamide and rosmarinic acid suggest that the effects of these compounds are partly related to their antioxidant activities. In three different experimental models of human neuroblastoma cell lines, i.e., SH-SY5Y cells exposed to oxidative stress, SK-N-BE cells treated with L-glutamate, and SH-SY5Y cells under conditions of hypoxia and reperfusion, the protective effects of clovamide and rosmarinic acid were found at their micromolar concentrations (EC_50_ from 0.9 to 3.7 µM). The neuroprotective effect of both compounds was comparable, reaching a 40 to 60% reduction in cell death [75].

Clovamides are also considered pharmacophore templates that could be useful in the treatment of neuroinflammation and neurodegeneration. Examinations of clovamide analogues with catechol functionality as potential anti-Parkinson’s disease agents have demonstrated that compounds with catechol groups exhibit better neuroprotective effects, with in vitro EC_50_ values ranging from 4.26 to 23.83 μM. The oral administration of the most potent derivative (signed as the compound 1) to rats (at a dose of 10 or 20 mg/kg) resulted in the alleviation of apoptosis and oxidative stress. The molecular mechanisms of this clovamide derivative action involved the PI3K/AKT/mTOR pathway-mediated upregulation of heme oxygenase-1 (HO-1) expression [76]. The PI3K/AKT/mTOR (the phosphoinositide 3 kinase (PI3K)/Akt/mammalian target of rapamycin (mTOR)) pathway is a major intracellular regulator of the cell cycle and the proliferation process. In the context of neuroprotective activity, this signaling cascade is an important regulator of the antioxidant and apoptosis-modulatory properties of the HO-1 enzyme. Furthermore, other work has revealed that clovamide-related compounds such as caffeoylquinic acids and phenylethanoid glycosides that contain two or more catechol moieties inhibit amyloid β-protein (Aβ) aggregation. A structure–activity relationship (SAR) study employing clovamide and a series of its derivatives, produced in reactions between L-dopamine, D-dopamine, L-tyrosine, or L-phenylalanine and caffeic acid, *p*-coumaric acid, or cinnamic acid, demonstrated the essential role of catechol moiety in the anti-aggregatory activity of the tested compounds. While the clovamide and clovamide-type compounds containing one or two catechol moieties were potent inhibitors of Aβ aggregation, the non-catechol-type derivative had little or no activity. The most effective compounds were D-clovamide (half maximal inhibitory concentration (IC_50_) = 1.6 µM), clovamide methyl ester (IC_50_ = 4.9 µM), L-clovamide (IC_50_ = 5.7 µM), and dihydroclovamide (IC_50_ = 8.7 µM) [77]. Recent studies have confirmed the importance of catechol moieties to the inhibitory action of clovamides on the human islet amyloid polypeptide (hIAPP) and the Aβ42/hIAPP disaggregation activity of these compounds [78]. The assumption regarding the important role of catechol moiety is also supported by studies on other polyphenols. For instance, a correlation between the presence and number of catechol moieties with anti-fibrillation activity was also found for A-type procyanidins and their derivatives [79].

Remarkably, an inhibitory effect on β amyloid aggregation was also evidenced in vitro and in vivo for rosmarinic acid, though direct interactions seem rather unlikely. On the other hand, results from studies on rosmarinic acid may partly reflect the absorption and pathways of the biological activity of the related compounds. The migration of polyphenolic substances into the brain is hindered due to the presence of the blood–brain barrier (BBB). Therefore, other mechanisms of the rosmarinic acid action, such as the stimulation of monoamine (norepinephrine, 3,4-dihydroxyphenylacetic acid, dopamine and L-dopamine) secretion, have also been postulated. It has been evidenced that dopamine and levodopa inhibit Aβ aggregation [80]. For that reason, an increase in the monoamine level in the brain after rosmarinic acid intake is most likely a consequence of the suppression of the Maob gene, which encodes the monoamine oxidase B (a dopamine-degrading enzyme) [81].

In opposition to the aforementioned findings indicating the essential role of catechol moiety in anti-aggregatory action, results from studies on other types of clovamide analogues suggest that the lack of these units does not lead to the loss of biological activity, including anti-inflammatory properties. It has been demonstrated that the dihydroxyl group of catechol moiety in caffeic acid residue is not essential for a reduction in nitric oxide synthesis under inflammatory conditions. Analogues of clovamide methyl ester, with the hydroxyl group of catechol moiety in caffeic acid and L-3,4-dihydroxyphenylalanine (L-DOPA) replaced by various functional groups, were found to be potent anti-inflammatory agents in BV2 cells. These synthetic clovamide analogues reduced the generation of NO production and the expression of the inducible nitric oxide synthase (iNOS). The most effective one was the 3,5-ditrifluoromethyl analogue (named as the 9l compound; IC_50_ = 2.8 μM), displaying about 26.3 times higher efficiency than the parent compound, i.e., clovamide methyl ester (IC_50_ = 73.6 μM) [82].

In another study, six synthetic clovamide analogues (with carboxylic acid or ester functionalities) were found to have significant inhibitory effects on iNOS activity in BV-2 cells, with IC_50_ values ranging from 1.01 to 29.23 μM. In Parkinson’s disease mouse models, the oral administration of the 4b analogue improved dyskinesia, reduced the expression of glial fibrillary acidic protein (GFAP; a marker of neuroinflammation) and increased a number of tyrosine hydroxylase-positive cells [83]. In BV-2 microglial cells, S-ethyl 2-oxopropanethioate (EOP), a synthetic clovamide derivative, was found to suppress the iNOS and cyclooxygenase 2 (COX-2) genes as well as diminish the synthesis of pro-inflammatory cytokines, mainly via the p38, the extracellular signal-regulated kinase (ERK) and NF-κB-dependent pathways [84]. In vitro and in vivo anti-neuroinflammatory effects were also evidenced for DPTP ([3-(3,4-dihydroxy-phenyl)-2-[4-(3-trifluoromethylphenyl)-but-2-enoylamino]-propionic acid methyl ester]), another synthetic clovamide derivative. DPTP blocked the cellular inflammatory response at different molecular levels, including a reduction in the nuclear factor of kappa light polypeptide gene enhancer in B-cells inhibitor alpha (IκBα) phosphorylation, resulting in the inhibition of NF-κB activation and the suppression of molecular pathways dependent on this modulator of gene transcription. Moreover, DPTP attenuated the phosphorylation of c-Jun N-terminal kinase (JNK), whose activation is a key element in stress signaling pathways as well as pathological cell death; for example, it is associated with neurodegenerative diseases [85]. In animals, a prophylactic treatment with DPTP (20 mg/kg) for 7 days reduced the glial activation and behavioral impairment induced by an intoxication with 1-methyl-4-phenyl-1,2,3,6-tetrahydropyridine (MPTP) [86].

### 4.4. Anti-Platelet Action

The antiplatelet effects of natural (poly)phenols have been demonstrated both in vitro and in dietary intervention studies [87,88]. Caffeic acid, a parental component of clovamide, was found to inhibit thrombin-induced platelet activation via the modulation of different intracellular pathways, including the up-regulation of the cyclic adenosine monophosphate (cAMP) level and inhibition of the protein kinase B (Akt) and ERK kinase [89]. However, the antiplatelet potential of clovamides seems to be disputable. The number of papers dealing with the effects of clovamide(s) on hemostasis is limited, and the available data are inconsistent. Caffedymine isolated from cocoa beans reduced the expression of P-selectin in platelets along with the COX inhibitory activity, suggesting the considerable antiplatelet potential of this compound [90]. The antiplatelet properties of clovamide, its derivatives and clovamide-rich extracts were described in 2006 by Park and Schoene [91], based on in vitro and animal studies. Clovamide and *N*-coumaroyldopamine inhibited the activation of platelets and their interactions with leukocytes. A decrease in the P-selectin (a marker of platelet activation) expression by about 30% was observed at a concentration of 0.05 μM (0.018 μg/mL) of these compounds. The fact that the inhibition was partly reduced by β_2_-adrenoceptor antagonists suggested that these receptors are probably involved in the anti-platelet effects of the examined clovamides. The β_2_-adrenoceptor–clovamide interactions may be based on a structural similarity between the clovamides and some ligands of β-adrenergic receptors, e.g., dobutamine and denopamine. In mice administered clovamide (50 and 100 µg per 35 g of b.w.), the platelet activation, including their interactions with leukocytes, was also reduced. 

On the other hand, more recent (2017) in vitro studies on *trans*-clovamide and clovamide-rich extracts isolated from aerial parts of three *Trifolium* species (i.e., *T. clypeatum*, *T. obscurum* and *T. squarrosum*) demonstrated only the moderate anti-platelet effects of the examined substances. At concentrations of 1–5 µg/mL, the reduction in platelet adhesion to the collagen or fibrinogen surface by clovamide or the extracts did not exceed 20%. Furthermore, no significant effects on platelet aggregation in platelet-rich plasma were found [30].

### 4.5. Anticancer Properties

The anticancer properties of phenolamides have been evidenced both in human and animal cell lines, and their action involves three main aspects that are essential for combating cancer cells, i.e., cell cycle arrest, pro-apoptotic effects and a reduction in metastasis (including the inhibition of cell migration and invasiveness) [5]. The amount of data related to the anti-cancer activities of clovamide itself is limited, but some information on the related compounds is available. The results of studies by Park and Schoene [92] revealed the ability of these compounds to inhibit the proliferation of cancer cells by *N*-coumaroyltyramine in U937 (human monoblastic leukemia cells) and Jurkat cells (human acute lymphoblastic leukemia T cell line). The suggested molecular mechanisms of this compound action involved an inhibition of the epidermal growth factor receptor (EGFR) tyrosine kinase, arrest of the cell cycle in the S phase and the induction of apoptosis. Recently, the enhancing effects of *N*-*trans*-*p*-coumaroyltyramine (TCT) on indomethacin and diclofenac-triggered cytotoxicity in a breast cancer cell line (MCF-7) have been also demonstrated. A combination of the above drugs with TCT significantly reduced the viability and the mitochondrial membrane potential of the examined cancer cells [93]. Furthermore, anti-proliferative and pro-apoptotic activity was found in studies on the *N*-feruloyltyramine-containing extract from goji fruits (*Lycium barbarum* L.). These experiments have provided the first data on the inhibitory effects of *L. barbarum*-derived extracts on the growth and proliferation of head neck cancer cells [94].

### 4.6. Antiviral, Antibacterial and Anti-Trypanosomal Activities

The available literature suggests the antimicrobial and antiviral properties of clovamides or clovamide-containing plant preparations. Clovamide and the clovamide-rich extract obtained from *Dichrostachys cinerea* (L.) Wight & Arn., a medicinal plant used in folk medicine against different infections, were preliminary examined in the context of their ability to combat the influenza A virus (H5N1) in vitro. Both the extract and clovamide inhibited cell infection by the influenza A virus (H5N1) in the experimental system of Madin–Darby Canine Kidney (MDCK) cells [19]. In another work on the antiviral efficiency of clovamide, in silico studies were performed to determine if clovamide-type compounds are able to inhibit the SARS-CoV-2 Main Protease (M^pro^). The M^pro^ is an enzyme that is directly involved in viral replication processes; therefore, the blockage of its activity may be one of the potential molecular mechanisms of the suppression of SARS-CoV-2 infection. Results from molecular docking, annealing-based molecular dynamics, and Density Functional Theory (DFT) calculations have suggested that clovamide and its derivatives may interact with catalytic and allosteric sites of the M^pro^ and have potential for further studies on their anti-SARS-CoV-2 activity [95].

It has also been suggested that clovamide may ameliorate *Helicobacter pylori* infections, being one of the main causes of gastritis and gastric ulcers. Studies on human adherent gastric adenocarcinoma epithelial cells demonstrated that clovamide partly inhibited the adherence of *H. pylori* [96]. In addition, the anti-parasitic activity of clovamide has been reported. Clovamide displayed a strong antitrypanosomal potential against *Typnosoma evansi* (IC_50_ = 3.27 μg/mL). For a reference trypanocidal drug, i.e., diminazene aceturate, the IC_50_ was of 0.72 μg/mL [19].

### 4.7. Estrogenic Activity

Many studies have confirmed the beneficial health effects of the phytoestrogens present in plant-based foods [97,98]. Phytoestrogens are also considered therapeutic alternatives to the pharmacological hormone therapy used to alleviate menopausal complaints. The estrogenic activity of most phytochemicals is an effect of their ability to interact with estrogen receptors (ERs). Although the term “phytoestrogens” is usually associated with isoflavones, other groups of polyphenolics can also display this effect [99]. However, in the case of clovamides, the literature provides only in silico predictions. *Cis*- and *trans*-clovamide were found to have an affinity for estrogen receptor α (ERα). Their docking energies (reflecting their ability to bind to the receptor) were established to be −119.8 and −113.6 kJ/mol, respectively. Moreover, these docking energies were more exothermic than those calculated for estradiol (−92.0 kJ/mol) and the phytoestogenic ligand genistein (−93.4 kJ/mol) [100].

## 5. Concluding Remarks

The biological activity of clovamide and its derivatives is still only partly recognized. Although the literature evidence indicates the multidirectional activity of these compounds, their contribution to overall human health has not been estimated yet. Based on the available data, the antioxidant, neuroprotective and anti-neuroinflammatory activities of clovamide(s) seem to be particularly promising directions in their research. On the other hand, despite very encouraging results from in vitro works, a limited number of in vivo reports blocks evaluation of the exact physiological relevance of clovamides action. Clovamide(s) activity definitely needs more advanced experimental models, including organoids and animal studies. Without the strong support of vivo results, clovamide activity at a systemic level remains only hypothetical. Another aspect is the bioavailability of clovamide(s), especially in the context of neuroprotective activity, and prospects for overcoming this obstacle. Contrary to other polyphenols, research on the development of clovamide delivery systems is poorly advanced. For example, in studies on rosmarinic acid, a clovamide analogue, numerous formulas have been developed to increase its local bioavailability, including rosmarinic-acid-loaded nanovesicles, niosomal gels, liposomes and ethosomes, nanoemulsion-based hydrogels and polyethylene glycol (PEG)ylated nanoparticles [101]. In the case of clovamides, no such data are available.

## Figures and Tables

**Figure 1 foods-13-01118-f001:**
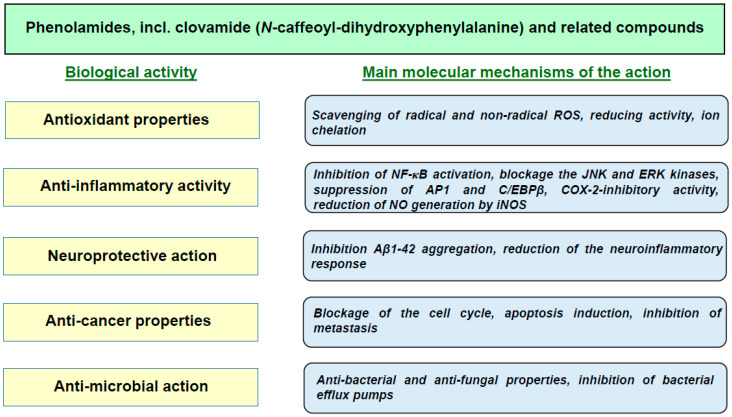
Main biological activities of phenolamides. Aβ1-42—amyloid peptide; AP-1—transcription factor activating protein 1; C/EBPβ—CCAAT/enhancer-binding protein; COX-2—cyclooxygenase-2; ERK—extracellular signal-regulated kinase; iNOS—inducible nitric oxide synthase; JNK—c-Jun N-terminal kinase; NF-κB—nuclear factor kappa-light-chain-enhancer of activated B cells; NO—nitric oxide; ROS—reactive oxygen species.

**Figure 2 foods-13-01118-f002:**
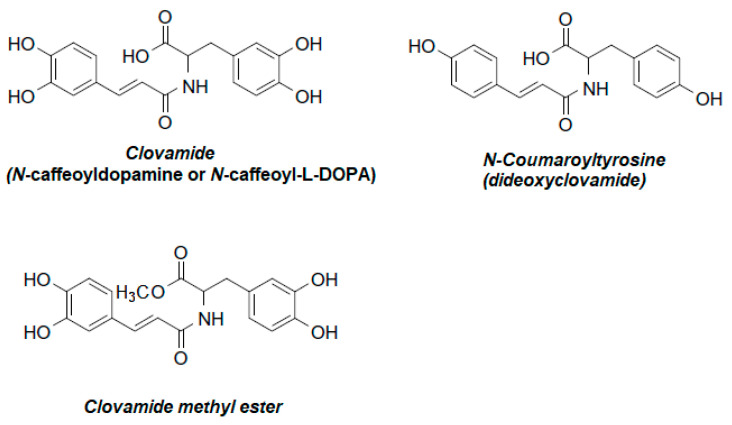
Exemplary clovamide-type compounds.

**Figure 3 foods-13-01118-f003:**
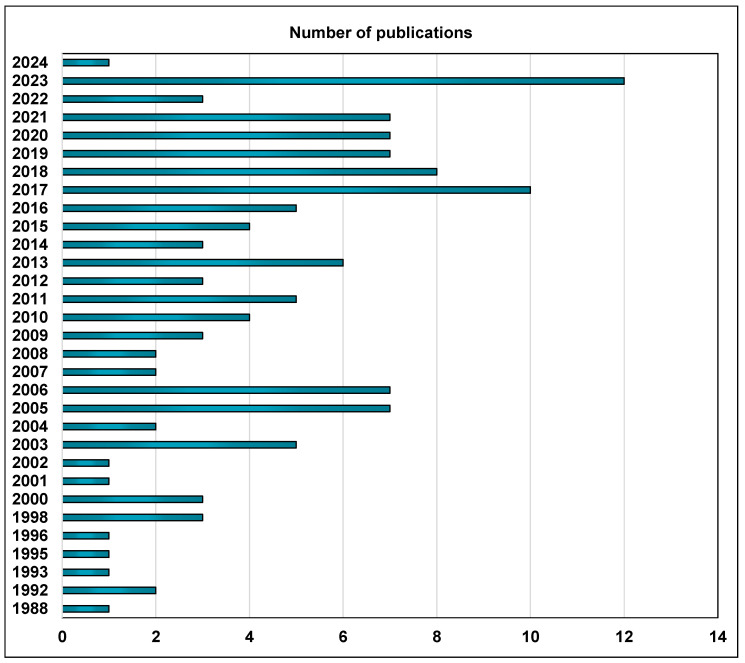
Publications related to clovamide(s). Data based on the Clarivate (Web of Science; access date: 28 March 2024.). The search criteria included the following: clovamide (all fields) or caffeoyl-*N*-tyrosine (all fields) or coumaroyl-*N*-tyrosine (all fields) or feruloyl-*N*-dopamine (all fields) or feruloyl-*N*-tyrosine (all fields) or *N*-caffeoyltyramine (all fields) *N*-feruloyltyramine (all fields) or deoxyclovamide (all fields) or dideoxyclovamide (all fields).

**Figure 4 foods-13-01118-f004:**
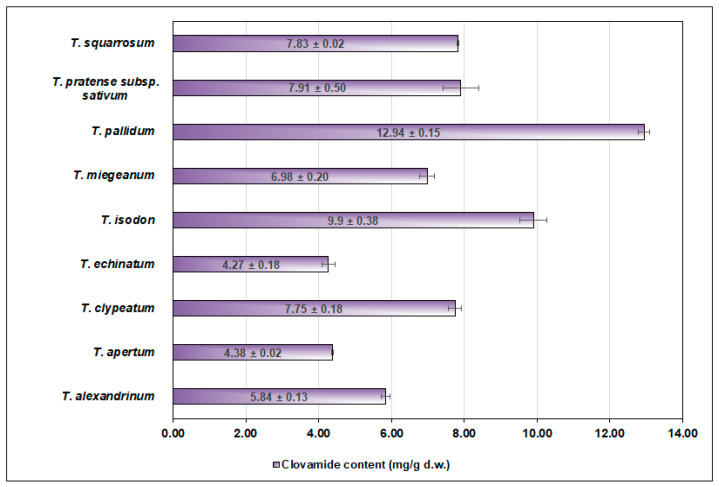
The content of clovamide in extracts from the aerial parts of various clover species. Plant material was extracted using 50% methanol, and the clovamide content was calculated based on the equivalents of chlorogenic acid [29].

**Table 1 foods-13-01118-t001:** The content of clovamide in fresh and processed cocoa beans and cocoa powder. In [23,24], the cocoa bean samples are defatted using dichloromethane and then extracted with methanol; in [25], the samples are defatted with petroleum ether and extracted using an internal standard solution in an ultrasonic bath.

Type of Sample	Clovamide Content	References
mg/g of the product		
Raw beans	0.052	[23]
Roasted beans	0.044	
Side products (winnowed)	0.024	
End products (winnowed)	0.065	
mg/g of cocoa powder		
Unroasted beans, of Ghana origin	0.0026	[24]
Roasted beans, of Ghana origin	0.0012	
Unroasted beans, of Arriba origin	0.0013	
Roasted beans, of Arriba origin	0.0005	
Unroasted beans, of Ivory Coast origin	0.0021	
Roasted beans, of Ivory Coast origin	0.0011	
mg/g in defatted raw beans		
18 samples of cocoa beans, originating from 12 countries, 4 continents	0.12–0.37	[25]

**Table 2 foods-13-01118-t002:** Clovamide content in relation to the total phenolics in methanolic (50%; *v*/*v*) extracts from the aerial parts of various clover species [29].

*Tifolium* Species	Clovamide % in the Total Phenolics Content
*T. alexandrinum* L.	21.54
*T. apertum* Bobrov	16.06
*T. clypeatum* L.	28.32
*T. echinatum* M.Bieb.	13.52
*T. isodon* Murb.	18.07
*T. miegeanum* Maire	14.20
*T. pallidum* Waldst. & Kit	36.90
*T. pratense* subsp. *sativum* (Schreb.) Ser.	19.61
*T. squarrosum* L.	15.12

## Data Availability

No new data were created or analyzed in this study. Data sharing is not applicable to this article.

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
