# Peer review of "Clovamide and Its Derivatives—Bioactive Components of Theobroma cacao and Other Plants in the Context of Human Health"

_foods, 2024, doi:10.3390/foods13071118_

Round 1
Reviewer 1 Report
Comments and Suggestions for Authors
This work summarized findings about clovamides, group of bioactive components of Theobroma cacao L. (cocoa) beans and other plants and available data on the biological activity of clovamide and its derivatives, in a context of their effects on human health. The article required a lot of work and careful data collection and analysis, the topic is well chosen and interesting, but I have the following objections:
A lot of medical terminology is used, which may be incomprehensible to readers of Foods who are outside from this field of research. Consider using simpler language. Also, the manuscript contains so many acronyms, which makes it even more difficult to follow. Try to avoid some of them.
Saccharomyces cerevisiae, in vivo, in vitro - write in italics.
Line 349-351 Are there any other examples of research on the anticancer activity of clovamide?
Line 372-377 Revise this paragraph to make it clearer.
Concluding remarks- this section should not contain a reference, the author should suggest a proposal for future research.
Author Response
This work summarized findings about clovamides, group of bioactive components of Theobroma cacao L. (cocoa) beans and other plants and available data on the biological activity of clovamide and its derivatives, in a context of their effects on human health. The article required a lot of work and careful data collection and analysis, the topic is well chosen and interesting, but I have the following objections:
A lot of medical terminology is used, which may be incomprehensible to readers of Foods who are outside from this field of research. Consider using simpler language. Also, the manuscript contains so many acronyms, which makes it even more difficult to follow. Try to avoid some of them.
Clovamides display a wide spectrum of biological activity, and in the future, some of them may be useful in a context of medicine. For that reason, different biomedical aspects and terms have appeared in this review. Due to an interdisciplinary character of the manuscript, it was very difficult to avoid specialist terminology and acronyms. However, following kind suggestions of the Reviewer, I have added more elucidations and explained the used acronyms, especially in sections devoted to molecular mechanisms of clovamide’s bioactivity (the 4.2. Anti-inflammatory effects and 4.3. Neuroprotective effects chapters). Some additional sentences have been also added to make the review easier to follow for readers representing different fields of science. Furthermore, the 4.6. Antiviral, antibacterial and anti-trypanosomal activities and 4.7. Estrogenic activity chapters have been rewritten and extended.
Saccharomyces cerevisiae, in vivo, in vitro - write in italics.
The suggested corrections have been done.
Line 349-351 Are there any other examples of research on the anticancer activity of clovamide?
Thank you very much for this suggestion. I have found two recently published works (Wongsakul et al., 2022 and Peraza-Labrador et al., 2022). The chapter has been extended by data published in these works.
Line 372-377 Revise this paragraph to make it clearer.
The paragraph has been revised and rewritten.
Concluding remarks- this section should not contain a reference, the author should suggest a proposal for future research.
During preparing the manuscript, I intended to present the conclusions in an opened form, i.e. rather as remarks and a prompt for undertaking studies and discussion, than as a strictly closed summary. For that reason, a reference has been added to present a huge differences in the advancement of research between clovamides and other polyphenols. In the revised version of the work, the main problems and future directions have been more clearly emphasized in this section.
Reviewer 2 Report
Comments and Suggestions for Authors
In the present manuscript "Clovamide and its derivatives – bioactives components of Theobroma cacao and other plants in a context of human health" the biological activity of clovamide and its derivatives are described. The review is very interesting, very well written, easy to understand, systematized, with tables and figures well represented. The bibliography is recent and correspond to the subject of the manuscript.
The manuscript corresponds to the requirements of the journal but I have some minor issues
- In the manuscripts there are tables with the content of clovamide, respectively with the content of total phenolics content in fresh and processed cocoa beans, cocoa powder, or aerial parts of clovers species. The question but also the suggestion would be- what kind of extracts were used, wthat solvents were used to extracts the compunds? In the case of clover specie, but also in the cacao bean samples the extracts were identical?
- Line 51 please correct the names in Latin of the species (italics)
Author Response
In the present manuscript "Clovamide and its derivatives – bioactives components of Theobroma cacao and other plants in a context of human health" the biological activity of clovamide and its derivatives are described. The review is very interesting, very well written, easy to understand, systematized, with tables and figures well represented. The bibliography is recent and correspond to the subject of the manuscript. The manuscript corresponds to the requirements of the journal but I have some minor issues
- In the manuscripts there are tables with the content of clovamide, respectively with the content of total phenolics content in fresh and processed cocoa beans, cocoa powder, or aerial parts of clovers species. The question but also the suggestion would be - what kind of extracts were used, what solvents were used to extracts the compounds? In the case of clover specie, but also in the cacao bean samples the extracts were identical?
Following kind suggestions of the Reviewer, I have added some more information on solvents used for extractions both in table heads and in the manuscript text. Data presented in the Table 2 derive from one, very extensive study, involving 57 clover species. All these extracts were obtained with the same solvent (50%MeOH).
In the case of the table 1 (clovamide content in cocoa beans), data derive from three works. Two of them (Arlorio et al., 2008 and Ye et al., 2021) contain information, that cocoa bean samples were treated with dichloromethane to remove the lipid fraction, and then extracted Soxhlet apparatus, using methanol.
In the third work (Lechtenberg et al., 2012), the authors provided following information “To 500 mg of the defatted powder 25.0 mL of the freshly prepared internal standard solution was added. Extraction was done in an ultrasonic bath”. No detailed information about the standard solution has been given.
- Line 51 please correct the names in Latin of the species (italics)
Italics have been applied in this line.
Reviewer 3 Report
Comments and Suggestions for Authors
The study presents an extremely interesting review of scientific knowledge about Clovamide and its derivatives.
However, the current version has a number of shortcomings that need to be clarified before a potential release.
- after reference 32 (L113) 33-34 are missing because in L122 it continues with references 41-44
- Table 2 is the extracted cluster 5 from the paper [29] and only partially because in that paper the mean values with the corresponding standard deviations are shown, and all three listed columns (in this paper) should also be shown with the corresponding standard deviations...although it would be best to show the results in the form of a graph, because then it would not be a mere rewriting of part of the published results, which in this paper form a separate whole
- table 3- 2nd column should contain the data "Clovamides efficiency" as EC50 or IC50 - however, according to the above - the column should not even be called "Clovamide (mg/ml)" because nothing else can be concluded from the data in the column, the same comment is for the values in the third column. This indicates that EC50 is the same as IC50, however there is a difference and the acronym 'I' in IC50, stands for inhibition, while 'E' in EC50, refers to effective. Therefore, it is necessary to modify table 3.
- in the last paragraph of the introduction (L61-72) it is stated that "recent decades have provided numerous reports dealing with beneficial properties of clovamide and related compounds", which is not supported by a single reference.
- I suggest to additionally make an overview in WoS of how many scientific publications there were on that topic, and even display them in the form of a graph.
- before the conclusions themselves, it would be interesting to get an insight into potential limitations and ideas in which direction the next research should be directed.
Sincerely
Author Response
The study presents an extremely interesting review of scientific knowledge about Clovamide and its derivatives. However, the current version has a number of shortcomings that need to be clarified before a potential release.
- after reference 32 (L113) 33-34 are missing because in L122 it continues with references 41-44
The mentioned references were related to a manuscript section devoted to clovamide(s) bioavailability. This part of the text has been accidentally deleted during formatting the FOODS journal template. In the revised version of the manuscript, both the missing fragment and references 33-40 have been correctly included.
- Table 2 is the extracted cluster 5 from the paper [29] and only partially because in that paper the mean values with the corresponding standard deviations are shown, and all three listed columns (in this paper) should also be shown with the corresponding standard deviations...although it would be best to show the results in the form of a graph, because then it would not be a mere rewriting of part of the published results, which in this paper form a separate whole
Thank you very much for this valuable suggestion. The Table 2 was prepared to present all data for different clovers in the most concise form. For that reason, it contained information directly extracted from the work of prof. Oleszek’s research team. Following kind suggestion of the Reviewer, I have modified this unsuccessful concept of the Table 2.
I have prepared the figure 4 containing data on the clovamide content in mg/g ±SD. Since the previous version of the Table 2 contained three parameters, it is very difficult to include them in the same graph. I tried to include both the clovamide content and the total phenolics content in the same graph, but due to the large differences in values related to the total phenolics and clovamide contents, the clovamide bars were very small and the standard deviation was tiny or invisible. Therefore, I have prepared a separate graph (Fig. 4 in the revised version of the manuscript) with clovamide content and a modified Table 2, providing only the clovamide % of the total phenolics content. Since the total phenolics content has been given without SD in the work of Oleszek et al. (2007), there was no possibility to provide it in the modified Table 2.
- table 3- 2nd column should contain the data "Clovamides efficiency" as EC50 or IC50 - however, according to the above - the column should not even be called "Clovamide (mg/ml)" because nothing else can be concluded from the data in the column, the same comment is for the values in the third column. This indicates that EC50 is the same as IC50, however there is a difference and the acronym 'I' in IC50, stands for inhibition, while 'E' in EC50, refers to effective. Therefore, it is necessary to modify table 3.
I strongly agree that differences between the EC50 and IC50 are obvious, however, both of them are used as parameters characterizing the activity of the examined compounds. Unfortunately, previous version of the Table 3 was misleading. The revised version of the Table 3 contains information what kind of parameters were calculated in the presented assays.
- in the last paragraph of the introduction (L61-72) it is stated that "recent decades have provided numerous reports dealing with beneficial properties of clovamide and related compounds", which is not supported by a single reference.
The mentioned sentence was a general statement to introduce the readers. All details and citations have been given thorough the manuscript body. In the revised version of the manuscript, to support this fragment of the Introduction section, I have slightly modified the sentence and added a new figure with publications based on the Web of Science (Fig. 3).
- I suggest to additionally make an overview in WoS of how many scientific publications there were on that topic, and even display them in the form of a graph.
Following suggestions of the Reviewer, I analyzed the WoS data and prepared a graph (the aforementioned Fig. 3) with numbers of publications related to clovamides.
- before the conclusions themselves, it would be interesting to get an insight into potential limitations and ideas in which direction the next research should be directed.
Some of problems and limitations such as poor data on bioavailability and small number of in vivo reports are presented in different chapters of the manuscript. Unfortunately, many of studies on clovamides have stuck at a stage of an in vitro tests. This is the main obstacle in evaluation of their physiological relevance. During preparing the manuscript, I intend to present the conclusion in an opened form - rather as remarks, than as strictly closed summary. In the revised version of the work, the main problems and future directions have been more clearly emphasized in the “Concluding remarks” section.
Round 2
Reviewer 3 Report
Comments and Suggestions for Authors
All corrections visible in the work are in accordance with my suggestions, and I have no further suggestions.
Sincerely